# Influence of Sociodemographic Determinants on the Hodgkin Lymphoma Baseline Characteristics in Long Survivors Patients Enrolled in the Prospective Phase 3 Trial AHL2011

**DOI:** 10.3390/cancers15010053

**Published:** 2022-12-22

**Authors:** Steeve Chevreux, Sandra de Barros, Camille Laurent, Amandine Durand, Cyrille Delpierre, Philippine Robert, Clémentine Joubert, Samuel Griolet, Salim Kanoun, Jean-Noël Bastie, René-Olivier Casasnovas, Cédric Rossi

**Affiliations:** 1Department of Hematology, Dijon-Bourgogne University Hospital, 21000 Dijon, France; 2INSERM, Unit 1231, University of Burgundy Franche-Comté, 21078 Dijon, France; 3Department of Hematology, University Institute of Cancer Toulouse-Oncopole, 31100 Toulouse, France; 4Department of Pathology, University Institute du Cancer, Toulouse University Hospital, 31300 Toulouse, France; 5INSERM, Unit 1037, Cancer Research Center of Toulouse-Purpan, Laboratoire D’excellence TOUCAN, 31037 Toulouse, France; 6Lymphoma Academic Research Organisation, CHU Lyon-Sud, 69495 Lyon, France; 7Nuclear Medicine Unit, Institut Universitaire du Cancer Toulouse-Oncopole, 31100 Toulouse, France; 8Department of Medicine, Divisions of Oncology and Hematology, Stanford University, Stanford, CA 94305-5458, USA

**Keywords:** Hodgkin lymphoma, social, PET, prognosis

## Abstract

**Simple Summary:**

While information about sociodemographic characteristics of Hodgkin lymphoma (HL) and their influence on outcomes remains elusive, the objective of this present study was to decipher the potential impact of social disparities on the disease features at diagnosis and to analyze how sociodemographic patient characteristics could impact HL outcomes. These findings suggest that some patient sociodemographic characteristics might impact access to medical care leading to a higher frequency of unfavorable presentations.

**Abstract:**

Introduction: Whereas numerous studies on several cancers describe the link between social conditions and disease severity, little is known about the social and demographic characteristics of Hodgkin lymphoma (HL) patients. At diagnosis, 10–15% of the patients in the advanced stages have a well-known poor outcome owing to their chemoresistance, but the determinants of the more advanced stages remain elusive. The objective of the present study was to decipher the potential impact of social disparities on the disease features at diagnosis and analyze how the sociodemographic patient features could impact the HL outcome of patients with advanced-stage HL enrolled in the AHL2011 trial. Methods: This ancillary study was conducted on a cohort of patients from French centers that had recruited more than five patients in the phase III AHL2011 study (NCT0135874). Patients had to be alive at the time of the ancillary study and had to have given their consent to answer the questionnaire. Pre-treatment data (age, gender, stage, B symptoms, IPS), the treatment received, the responses to PET-CT, and the presence of serious adverse events (serious adverse events—SAEs) were all extracted from the AHL2011 trial database. Sociodemographic data—marital status, living area, level of education, socio-professional category, and professional situation—were extracted from the questionnaires. The population density at the point of diagnosis was determined based on ZIP Code, and the distance from the reference medical center was then calculated by the road network. Baseline PET acquisition was performed before any treatment. PET images at baseline were centrally reviewed. The total metabolic tumor volume (TMTV) at the baseline was calculated using a 41% SUVmax cutoff for each lesion. Progression-free survival was defined as the time from randomization to the first progression, relapse, or death from any cause or the last follow-up. The data cutoff for the analyses presented here was 31 October 2017. The progression-free survival was analyzed on an intention-to-treat basis. Results: Among the 823 patients enrolled in the AHL2011 study, the questionnaire was sent to 394 patients, of whom 232 (58.9%) responded. At the time of HL diagnosis, 61.9% (N = 143) of patients declared that they were not socially isolated, 38.1% (N = 88) that they were single, 163 (71.2%) had a professional activity, and 66 (28.8%) were inactive owing to unemployment, retirement, or sick leave. Of the patients, 31.1% (N = 71) lived in a rural region, compared to 68.9% (N = 157) that lived in an urban region. The residence ZIP Code at the time of HL diagnosis was available for 163 (70%). Sociodemographic characteristics did not influence the presence of usual prognostic factors (ECOG, B symptoms, bulky mass, IPS) except for professional activity, which was associated with more frequent low IPS (0–2) (79 (48.5%) active versus 20 (30.3%) inactive patients; *p* = 0.012). Likewise, no correlation was observed between TMTV and sociodemographic characteristics. However, the TMTV quartile distribution was different according to the living area, with the two upper quartiles being enriched with patients living in a rural area (*p* = 0.008). Moreover, a negative correlation between the average number of the living area’s inhabitants and TMTV (R Pearson = −0.29, *p* = 0.0004) was observed. Conclusion: This study focused on sociodemographic parameters in advanced-stage HL patients and shows that professional activity is associated with more favorable disease features (low IPS), while patients living in rural or low-populated areas are more likely to have an unfavorable HL presentation with a high tumor burden (high TMTV). These data suggest that some patient sociodemographic characteristics might impact either access to medical care or environmental exposure, leading to a higher frequency of unfavorable presentations. Further prospective sociodemographic studies are necessary to confirm these preliminary results.

## 1. Introduction

Approximately 8500 new patients (3710 females and 4790 males) are diagnosed with HL every year in the US, and 2127 new cases of HL are identified per year in France, leading to an annual incidence rate of 3.7 and 2.7 per 100,000 person-years [1]. HL is one of the most curable cancers, with a successful treatment rate of 75% worldwide. However, there are differences in patient survival related to the disease stage at diagnosis, with earlier stages being associated with better survival outcomes. According to the Surveillance, Epidemiology, and End Results (SEER) database, between 1998 and 2014, the five-year survival rate for patients with localized HL was 15% greater than for those with an advanced stage [2]. The stage of diagnosis is also important as it drives the treatment strategy. Patients with early-stage HL usually receive abbreviated courses of chemotherapy followed by radiation therapy (combined modality), while those with an advanced-stage disease receive a more prolonged and eventually dosed-dense combination chemotherapy, while the baseline total metabolic tumor volume (TMTV) assessed by positron-emission tomography (PET) is an independent prognostic factor [3,4]. The patient outcome prediction at the baseline remains unsatisfactory. One challenge is diagnosing the disease as early as possible to maximize limited rather than advanced stages, but no effective screening test or early detection method is currently available, and patients usually display no specific symptoms (palpable lymph nodes, fever, night sweats, weight loss). As previously reported in the USA, early access to healthcare services can be beneficial in detecting early symptoms, which leads to diagnosing the disease at an earlier stage, which is associated with a better outcome. Indeed, several studies have found an association between the lack of adequate health insurance and advanced cancer stages at diagnosis [5,6,7]. Therefore, patients who are uninsured or have public insurance have worse HL-specific survival rates compared to those with private or military insurance [8]. Adults who lack health insurance are more likely to delay or skip medical care due to financial issues and, as a result, have an increased risk of poorer outcomes [9]. In France, a recent study has pinpointed some factors to explain social disparities (income, place of residence, level of information on the subject, isolated status) [10].

Among the socio-psychological parameters, marital status appeared to be correlated with survival, with isolated status conferring a more unfavorable HL outcome [11]. Overall, 5-year survival was 37.3% for widowers versus 80.9% for married patients (*p* < 0.001). The place of residence seems to be linked to differences in cancer care. Indeed, a lower medical density outside of big cities and the intrinsic social characteristics of the inhabitants (lower education level, lower income) were reported to explain the delay in diagnosing cancer [12]. In colorectal and cervical cancers, geographic disparities were associated with a higher risk of mortality in rural patients. In this study, a lower percentage of rural patients (65%) had access to a medical oncologist within 30 min of their residence compared to urban patients (94%) [13]. An Italian meta-analysis calculated that a distance greater than 50 miles between the patient and their oncologist was correlated with a poor prognosis and a more advanced stage for several cancers, including diffuse large B-cell lymphoma [14]. A French study has highlighted that the differences in local facilities, including the presence or not of an expert center in the field, could impact lymphoma outcomes [15]. Lymphoma care management in expert centers provides a better chance of disease-free survival related to better physician training, a multidisciplinary approach to patient management, and better access to clinical trials. The accessibility of expert care management is, therefore, a crucial issue. 

Therefore, we analyzed the impact of the sociodemographic patients’ characteristics and the accessibility of healthcare on advanced HL patients enrolled in the AHL2011 phase III trial [16,17]. This study contributed to establishing a PET-tailored strategy with a better benefit/risk ratio, decreasing acute and late toxicities without impairing tumor control, and a better cost/benefit compared to alternative strategies [18]. A model was developed using a 20-year time projection to compare different treatment strategies in advanced-stage HL, and the PET-guided AHL2011 strategy has the best profile in terms of efficacy and toxicity (studied using QALY data) but also at the medico-economic level [18]. Finally, a baseline TMTV with a threshold of 220 cm3 was shown to be predictive of patient outcome in the AHL2011 study [4]. 

The objective of this present study was to decipher the potential impact of social disparities on the disease features at diagnosis, especially regarding HL stage and TMTV, and to analyze how sociodemographic patient characteristics could impact HL outcomes. 

## 2. Material and Methods

### 2.1. Patients and Study Design

This ancillary study was conducted on a cohort of patients from the prospective study AHL2011 (NCT01358747) [16,17] which enrolled patients aged 16 to 60 years who had Ann Arbor stage III, IV, or IIB with a mediastinum-to-thorax ≥ 0.33 or extranodal localization. The complete eligibility criteria and treatment strategies tailored by interim PET are presented in the AHL2011 trial [16]. The patients (N = 823) were randomly assigned to a standard arm (N = 413) or a PET-guided arm (N = 410) from 19 May 2011 to 29 April 2014.

Only patients coming from French centers that recruited more than five patients into AHL2011 were eligible to optimize the logistical issues with a reasonable number of centers involved. The patients had to be alive at the time of the ancillary study and had to have given their consent to answer the questionnaire. This retrospective study was conducted in accordance with the Declaration of Helsinki and was authorized by the Eastern France Ethics Committee (2011/01—EudraCT N°: 2010-022844-19), allowing the computerized management of the medical data. The participants were informed of the research purposes and had a right of opposition.

Pre-treatment data (age, gender, stage, B symptoms, IPS), the treatment received, the responses to PET-CT, and the presence of serious adverse events (serious adverse events—SAE) were all extracted from the AHL2011 trial database.

### 2.2. Socio-Demographical Data

Sociodemographic data (marital status, living area, level of education, socio-professional category (CSP), and professional situation) were compiled by the clinical research team of the medical and clinical pharmacology department of the Toulouse University Hospital. In the questionnaire, to take account of the young age of the trial, the marital status was explored in terms of a partner in the same residency (family, boy/girlfriend, spouse), and not restricted to married status only.

The population density in the area of the patient’s place of residence at the time of diagnosis was determined on the basis of a ZIP code, and subsequently, the distance from the reference medical center was then calculated using the road network. The estimation of the inhabitant’s number and urban/rural categories were determined according to the definition of the INSEE (French National Institute of Statistics and Economic Studies [19]). 

The delay considered for initiating treatment was calculated as the time between the biopsy and the first day of treatment (C1d1).

### 2.3. PET/CT Acquisition and Analysis

The baseline PET acquisition was performed before any treatment, and the quality criteria required were previously detailed [16]. 

The PET images at baseline were centrally reviewed by three readers (S.K., A.S.C., and M.M.) blinded to the medical information, and were analyzed using the free open-source software, Beth Israel Plugin for Fiji (http://petctviewer.org, accessed on 2 February 2021).

The pathological uptake was defined by an increased uptake of 18-FDG over the physiological background. The total metabolic tumor volume (TMTV) at the baseline was calculated using a 41% SUVmax cutoff for each lesion [20]. In this study, all PET2 responses were centrally evaluated using the Deauville score (DS) [21], and PET positivity was defined according to the criteria used in the AHL2011 study, which are considered more reproducible with better positive predictive values than the classic DS. Indeed, interim PET with a DS 4–5 and SUVmax of the residual mass greater than 140% of the liver background was considered positive in the AHL study based on previous data showing better reproducibility and accuracy of this threshold compared to a visual analysis [22].

### 2.4. Statistics

We assessed the efficacy of the treatment in terms of interim PET response and progression-free survival. The delay considered for initiating treatment was calculated as the time between the biopsy and the first day of treatment (C1d1). The progression-free survival was defined as the time from randomization to the first progression, relapse, or death from any cause or the last follow-up. The data cutoff for the analyses presented here was 31 October 2017. The progression-free survival was analyzed on an intention-to-treat basis. Survival estimates with 95% confidence intervals (CIs) were calculated with the Kaplan-Meier method. The survival distributions were compared with stratified log-rank tests, and Cox proportional hazard regression models were used to estimate HRs and associated 95% CIs. Multivariate analyses were conducted using a Cox proportional hazard model. 

Differences between groups were significant if *p*-values were less than 0.05. The population characteristics were compared using Fisher’s exact test or the X^2^ test for discrete variables and the Kruskal-Wallis or Wilcoxon test for continuous variables. 

All analyses were produced with the SAS software (version 9.3).

## 3. Results

### 3.1. Patients

The questionnaire was sent to 394 of the 823 patients enrolled in the AHL2011 study, of whom 232 (58.9%) responded (Figure 1). Among the 232 patients who completed the survey, the residence ZIP code was available for 163 (70%) patients. The disease characteristics were comparable to those of the whole AHL2011 study (Table 1), although an enrichment in stage IIB was observed among patients with a known living area (*p* = 0.013). The median age was 32 years (16–60), ECOG was 0–1 in 217 patients (93.9%), B symptoms were present in 166 (71.6%) patients, and 60 (25.9%) and 136 (58.6%) patients had Ann-Arbor stages III and IV, respectively. The PET images were centrally reviewed, and the TMTV was calculated for 216 patients with a median value of 206 cm^3^ (18–1343). The PET response was assessed as negative in 205 (90.3%) and 206 (94.9%) patients after 2 and 4 cycles, respectively. The SAEs were reported in 74 (31.9%) patients.

### 3.2. Analyses of Sociodemographic Characteristics

Out of 232 patients, 143 (61.9%) declared that they were not socially isolated (i.e., married or in a married life relationship) and 38.1% (N = 88) that they were single (divorced, separated, widowed, student alone), only two of whom were widowed, i.e., 0.9% of the patients. At diagnosis, the non-isolated patients were older, with a median age of 38.7 years versus 28.7 years for isolated patients (*p* < 0.001).

Among the patients, 71 (31.1%) lived in a rural region, compared to 157 (68.9%) in an urban region.

A total of 131 (58%) patients declared an educational level lower than that of a high school graduate and 95 (42%) a higher level. The socio-professional categories were: farmer (N = 5; 2.2%), (N = 15; 6.5%) craftsmen/tradesmen, manager (N = 40; 17.2%), intermediate profession (N = 13; 5.6%), employee (N = 99; 42.7%), worker (N = 27; 11.6%), student (N = 11: 4.7%), and unstable professional activity (N = 15; 6.5%). Therefore, farmers/artisans/workers/others represent 33.3% (N = 75) of the patients, and managers/employees/students represent 66.7% (N = 150). Overall, 163 patients (71.2%) had professional activity, and 66 (28.8%) were inactive owing to unemployment (N = 24), retirement (N = 3), or sick leave (N = 39). 

### 3.3. Relationships between Sociodemographic Features and Disease Characteristics

Sociodemographic characteristics were well balanced according to the presence or not of usual prognostic factors (ECOG, B symptoms, Bulky Mass, IPS). A total of 79 (48.5%) of the 163 patients with a professional activity and 20 (30.3%) of the 66 inactive patients had low IPS (0–2) (*p* = 0.012). Specifically for the high IPS and inactive group (N = 9), the link is no longer significant when we exclude sick leave patients (*p* = 0.144), but 18/102 (18%) of patients with high IPS had no professional activity vs. 9/88 (10%) patients with low IPS.

No correlation was observed between the TMTV and the usual prognostic factors or sociodemographic characteristics. However, the TMTV quartile distribution was different according to the living area as analyzed in the subset of patients with available residence ZIP codes, with the two upper quartiles being enriched with patients living in rural areas (*p* = 0.008). Moreover, a negative correlation between the living area’s number of inhabitants and TMTV (R Pearson = −0.29, *p* = 0.0004) was observed (Figure 2). A TMTV < 220 cm^3^ was more frequent (67.8%) in patients living in areas with a high median density (1228 inhabitants/km^2^) compared to those (63.8%) living in areas with a lower median density (943 inhabitants/km^2^) (*p* = 0.038). Conversely, the distance between the patient’s residence and the reference center had no impact on TMTV: the median distance was of 38.7 km and 45.6 km for patients with low and high TMTV, respectively (*p* = 0.205).

### 3.4. Relationships between Sociodemographic Features and PET Responses

Among the patients, 22 (9.7%) and 11 (5.1%) had a positive PET after 2 and 4 cycles, respectively. In the whole cohort, interim PET positivity was associated with reduced PFS estimates in univariate analysis (5-year PFS = 92.3% in PET2-PET4-, 75.4% [HR = 3.26; 95%CI: 1.84–5.78] in PET2+/PET4−, and 46.5% [HR = 11.95; 95%CI: 7.32–19.52] in PET4+ patients, respectively; *p* < 0.0001). No significant difference in the PET response rate was observed according to marital status, professional activity, educational graduate level, or living area. 

### 3.5. Relationships of Sociodemographic Features and Outcome of These Long Survivor Patients

With a median follow-up of 5.7 years in these 232 patients (CI 5.6–5.9), a total of 27 PFS events occurred versus 17 in the subset of 163 patients with a similar median follow-up of 5.7 years (CI 5.5–5.9). The median progression-free survival and overall survival were not reached in the whole cohort or either subset with the current follow-up. Overall, the 5-year PFS was 88.8% (95% CI 84–92.2) among the 232 patients who completed the survey and 90.2% (95% CI 84.5–93.9) in the subset with living area information available. The PFS was similar when comparing these sociodemographic categories side by side, as reported in Table 2. Of note, SAEs were less frequent (28.2% (N = 46)) in active than in inactive patients (42.4% (N = 28); *p* = 0.037).

## 4. Discussion

Whereas the social aspects have scarcely been reported in the literature to date, this ancillary study of the prospective phase III AHL2011 trial focuses on the analyses of the sociodemographic characteristics at the time of the diagnosis of advanced HL in patients aged 15–60. 

First, professional activity was associated with a lower prognostic score IPS and significantly less frequent SAE. We can therefore speculate that there is a more proactive response to symptoms in this active population compared with inactive patients, in both the disease onset and treatment periods. Based on the more frequent SAEs reported in patients with no professional activity, we can propose reinforcing the care management for those patients, specifically in the intercourse period of intensive chemotherapy. To do that, the role of general practitioners is crucial across the cancer trajectory [23].

Second, the socially isolated patients were younger, which is consistent with the trend of rising age at marriage in Western countries [24]. High TMTV, which is reported as an unfavorable independent prognostic factor in HL [4], was associated with living area. Indeed, patients living in rural and smaller towns had higher TMTV. This significant correlation remains mitigated (R = 0.29) insofar as the numbers of inhabitants in these towns were heterogeneous. To explain this association, some occupational insecticide exposure could be stronger in agricultural areas close to residential properties [25]. The potential role of such exposure in the onset and progression of lymphomas has been reported [26]. Moreover, these higher tumor masses in smaller towns could stem from the medical care network in France, with a lower density of medical support in the rural area, leading to a longer lead time to detect pathological symptoms and obtain a diagnosis than urban patients. Therefore, this long delay could explain the higher TMTV associated with living areas. Similar results were observed in other cancers, particularly in breast cancer patients [12]. Surprisingly, the distance between the referent center and the living area was not correlated with survival, whereas this association was previously pinpointed in a French epidemiological study [15] and in a meta-analysis including lymphomas [14], in which a greater distance of 50 miles between the patient and their oncologist was correlated with a worse prognosis and a more frequent advanced stage. This discrepancy with our analysis could reflect the large disparity in the multiple participating centers (i.e., towns) in the AHL2011 trial, resulting in a diluted impact of the distance. Indeed, as mentioned above, the correlation between the TMTV and the number of inhabitants in the living area seems more relevant in our study than differences in local facilities. This significant correlation remains mitigated (R = 0.29) insofar as the numbers of inhabitants in these towns were heterogeneous.

The sociodemographic characteristics did not significantly impact early response to treatment or the patient outcome in terms of PFS, while an unfavorable prognostic impact of PET2/4 positivity was similar in this series and the whole AHL2011 cohort [17]. We can hypothesize that the small number of relapses hampered the identification of any prognosis impact. Moreover, patients with the more severe disease who died of HL were not enrolled, hampering conclusions on the statistical correlation between social parameters and outcome. The isolated status did not influence the outcome in this cohort of young patients, whereas Wang et al. [27] reported in an elderly HL population that widowed patients had a significantly higher risk of mortality related to psychological factors. Here, our study included only 2 widowed patients, and overall, better social and care support for younger patients may explain the lack of correlation with overall survival. 

The pitfalls of the study are mostly linked to the retrospective analysis of sociodemographic features, which leads to limiting the study to live patients who agree to answer the questionnaire, obviously inducing some bias. However, the studied population seems representative of the whole AHL2011 cohort, and although we selected centers that had the highest experience in patient management within the AHL2011 study to collect social data, the patients’ characteristics in the present series and the whole cohort are similar.

Altogether, these results prompt us to pay more attention to sociodemographic parameters to improve medical care at an individual and territorial level. Indeed, in future clinical trials or in prospective real-life studies, such as the REALYSA study conducted in France [28], patients’ sociodemographic characteristics, including their educational level and living area, deserve to be explored to better understand the reasons for determining the unfavorable presentation of the disease, such as a high TMTV or IPS. 

## 5. Conclusions

To the best of our knowledge, this is one of the first studies focused on the detailed social demographic parameters of HL patients. Based on these results, the outlook could be to implement dedicated prospective surveys on social items in future HL clinical trials and enlarge the field to all cancers to improve patients’ medical care and management.

## Figures and Tables

**Figure 1 cancers-15-00053-f001:**
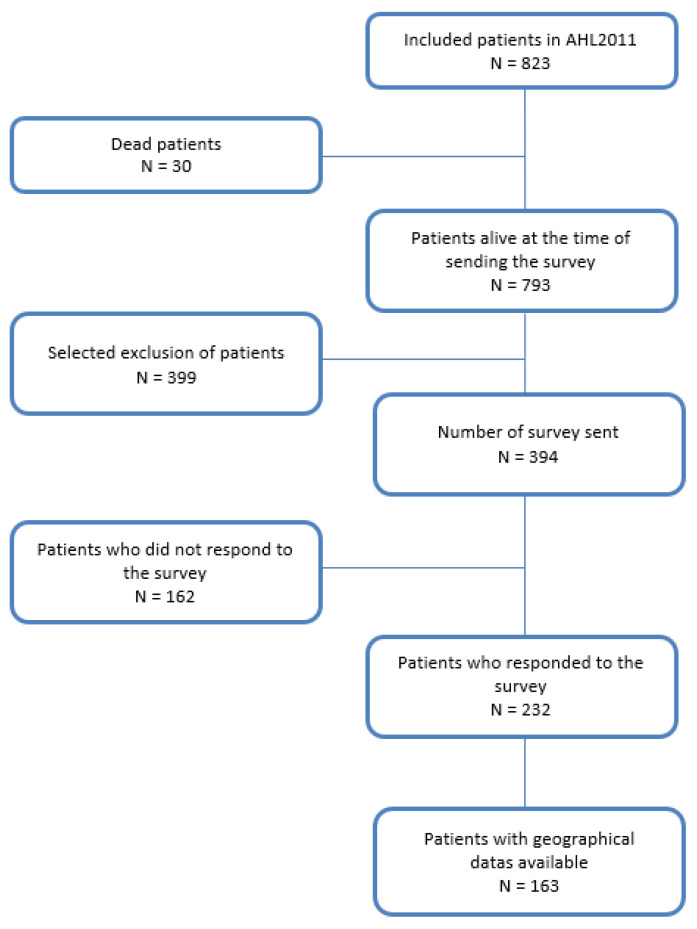
Flowchart displaying the number of patients who completed the survey and those with the ZIP codes reported (geographical data available) in the AHL2011 trial.

**Figure 2 cancers-15-00053-f002:**
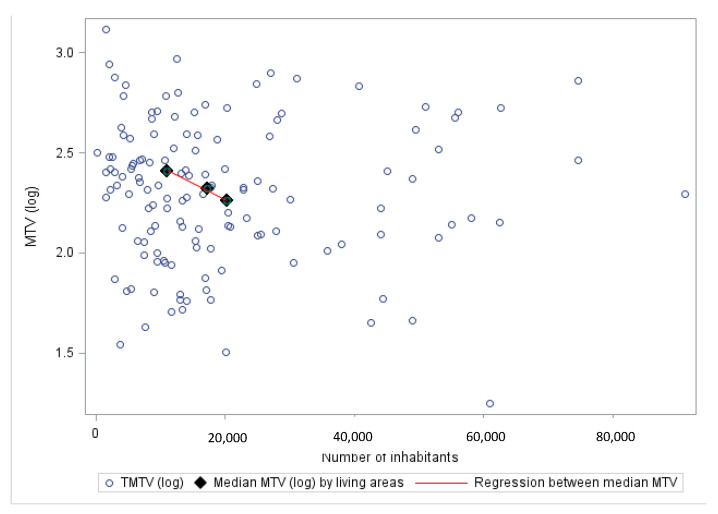
Correlation between the TMTV and the number of inhabitants in the town of patients who provided their ZIP codes. All sizes of towns are considered in the calculation, while for the visualization, the upper limit of the x-axis is 80,000. Abbrev: MTV. Metabolic Tumor Volume.

**Table 1 cancers-15-00053-t001:** Key baseline patient characteristics with the respective subsets according to the availability of the questionnaire.

Patients’ Characteristics	AHL2011 Whole Cohort	Completed Survey		ZIP Code Available	
	N = 823	N = 232	Test	N = 163	Test
Median age. years (range)	30 (16–60)	32 (16–60)	Wilcoxon *p* = 0.002	32 (17–59)	Wilcoxon *p* = 0.018
Male. No. (%)	516 (62.7%)	141 (60.8%)	Chi-2 *p* = 0.475	100 (61.3%)	Chi-2 *p* = 0.691
ECOG. No. (%)			Chi-2 *p* = 0.766		Chi-2 *p* = 0.839
0	396 (48.4%)	112 (48.5%)		76 (46.9%)	
1	365 (44.6%)	105 (45.5%)		73 (45.1%)	
2	58 (7.1%)	14 (6.1%)		13 (8.0%)	
B symptoms. No. (%)	263 (32.0%)	66 (28.4%)	Chi-2 *p* = 0.176	48 (29.4%)	Chi-2 *p* = 0.443
Ann Arbor stage. No. (%)			Chi-2 *p* = 0.139		Chi-2 *p* = 0.092
I or II	98 (11.9%)	36 (15.5%)		28 (17.2%)	
III	229 (27.8%)	60 (25.9%)		43 (26.4%)	
IV	496 (60.3%)	136 (58.6%)		92 (56.4%)	
Stade IIB	87 (10.6%)	32 (13.8%)	Chi-2 *p* = 0.060	26 (16%)	Chi-2 *p* = 0.013
Bulky Mass. No. (%). cm					Chi-2 *p* = 0.695
≤10	462 (62.5%)	138 (63.9%)	Chi-2 *p* = 0.62	99 (63.9%)	
>10	277 (37.5%)	78 (36.1%)		56 (36.1%)	
IPS group. No. (%)			Chi-2 *p* = 0.559		Chi-2 *p* = 0.409
0–2	343 (41.9%)	101 (43.5%)		73 (44.8%)	
≥3	475 (58.1%)	131 (56.5%)		90 (55.2%)	
PET2 central review. No. (%)			Chi-2 *p* = 0.121		Chi-2 *p* = 0.162
Positive	100 (12.6%)	22 (9.7%)		15 (9.3%)	
Negative	695 (87.4%)	205 (90.3%)		146 (90.7%)	
PET4 central review. No. (%)					Chi-2 *p* = 0.932
Positive	43 (5.7%)	11 (5.1%)	Chi-2 *p* = 0.653	9 (5.8%)	
Negative	716 (94.3%)	206 (94.9%)		146 (94.2%)	
TMTV0 class					Chi-2 *p* = 0.994
< 220 cm^3^	386 (52.2%)	118 (54.6%)	Chi-2 *p* = 0.402	81 (52.3%)	
≥ 220 cm^3^	353 (47.8%)	98 (45.4%)		74 (47.7%)	

Abbreviations: ECOG, Eastern Cooperative Oncology Group; IPS, international prognosis score; PET. positron emission tomography; TMTV, total metabolic tumor volume.

**Table 2 cancers-15-00053-t002:** Survey response and progression-free survival after 5 years according to the sociodemographic characteristics of patients who completed the survey.

	Frequence N(%)	PFS 5 Years
Marital status		
Isolated	143 (61.9%)	92 %
Non-isolated	88 (38.1%)	86 %
Living area		
Rural	71 (31.1%)	90.1%
City	157 (68.9%)	87.3%
Diploma		
High school graduation or below	131 (58.0%)	88.5%
Above high school graduation	95 (42.0%)	88.4%
Socio-professional categories		
Farmers/artisan/workers	75 (33.3%)	90.7%
Executives/employees/student	150 (66.7%)	87.3%
Professional context		
Active	163 (71.2%)	89.4%
Non active	66 (28.8%)	87.7%

## Data Availability

The original data cannot be shared due to limited access to the databases.

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
