# Peer review of "Influence of Sociodemographic Determinants on the Hodgkin Lymphoma Baseline Characteristics in Long Survivors Patients Enrolled in the Prospective Phase 3 Trial AHL2011"

_cancers, 2022, doi:10.3390/cancers15010053_

Round 1
Reviewer 1 Report
Very interesting paper concerning the correlation between socio-demographic characteristics and outcome in HL patinets enrolled in in the AHL2011 trial. Of great interst in the conclusions the consideration about rural patients and possible occupational insecticide exposure as well as to slower access to medical assistance . no ciorrelation was found probbaly beacsuse of the high curability of the disease with low relpase rate
Tw comments:
1) a stated by the authors , there are the typical biases of a retrospectuive trial, first of all the fact the dead patients could not answered to the questionnaire
2) More than 40% of the patients di not answered to the questionnaire. Despite it could be predictable, this is is a high percentage and results coukl be quite different. This item should be mentioned in the discussion
Author Response
Reviewer 1
Comments and Suggestions for Authors
Very interesting paper concerning the correlation between socio-demographic characteristics and outcome in HL patients enrolled in in the AHL2011 trial. Of great interst in the conclusions the consideration about rural patients and possible occupational insecticide exposure as well as to slower access to medical assistance . no correlation was found probably because of the high curability of the disease with low relapse rate
Two comments:
- As stated by the authors, there are the typical biases of a retrospective trial, first of all, the fact the dead patients could not answer the questionnaire
We thank the reviewer for this remark. there are indeed pitfalls due to the retrospective aspect of the study leading to exclude the dead patients. But as discussed and presented in table 1, the patients’ characteristics are not different between the whole cohort and the subset of patients studied in this ancillary analysis, suggesting that the present study has limited bias regarding patients selection within the AHL2011 study.
2) More than 40% of the patients did not answer the questionnaire. Despite it could be predictable, this is a high percentage and results could be quite different. This item should be mentioned in the discussion
Indeed, 41% of patients did not answer the questionnaire which is unfortunately consistent with the literature for this kind of ancillary study not previously planned at the launch of the clinical trial. The lead time after the end of the study could also partly explain the low rate of reply.
This point is now mentioned in the discussion: ‘Despite our careful efforts to mitigate the lack of response, 41% of patients did not answer the questionnaire as observed in a similar post-hoc study (SIMONAL study) in which 50% of patients did not answer a questionnaire dedicated to the fatigue assessment (Mounier et al, Cancer, 2019). In the present work, we could also speculate that the long lead time after the end of the clinical trial could lead to a decreased patient interest and motivation for filling out the questionnaire.’
Reviewer 2 Report
The present manuscript is a retrospective study that aimed to associate some socio-demographic characteristics of HL and prognosis and outcomes in advanced HL enrolled in the AHL2011 trial, one of the most recent randomized trials on PET-adapted strategies in advanced HL. While this paper covers a relevant research question, the manuscript needs major revision, and a new version of the article should be provided.
Major revision
1-A large number of studies have documented the impact of socioeconomic status inequalities on cancer survival, even in highly developed countries. The literature concerning only Hodgkin Lymphoma is not so extensive. Still, the authors could improve the Introduction and Discussion by adding references about socioeconomic status and prognosis/outcome in Hodgkin Lymphoma and NHL.
2- The study's objective in the abstract differs from the objective in the introduction.
3- The retrospective nature of the study and missing information from pts dead at the moment of the analysis prevents a proper analysis of the impact of socio-demographic characteristics on the outcome because this is a critical selection bias. The authors have expressed their concern about this issue in the discussion and argued that the studied population seems to be representative of the AHL2011 cohort, but this is not sufficient to conclude that it is possible to perform an outcome analysis. Therefore, the authors should consider restricting the study to the impact of characteristics on baseline characteristics at diagnosis and prognostic factors, but not the outcome. Another possibility to pursue the outcome analysis is to retrospectively collect occupational position and use it as a proxy of socioeconomic status in patients who died, even though the remaining information from the survey will not be available. This measure is one of the most commonly used indicators of socioeconomic status (Silvia Stringhini*, Cristian Carmeli*, Markus Jokela Published online January 31, 2017 http://dx.doi.org/10.1016/S0140-6736(16)32380-7
Author Response
Reviewer 2
The present manuscript is a retrospective study that aimed to associate some socio-demographic characteristics of HL and prognosis and outcomes in advanced HL enrolled in the AHL2011 trial, one of the most recent randomized trials on PET-adapted strategies in advanced HL. While this paper covers a relevant research question, the manuscript needs major revision, and a new version of the article should be provided.
Major revision
1-A large number of studies have documented the impact of socioeconomic status inequalities on cancer survival, even in highly developed countries. The literature concerning only Hodgkin Lymphoma is not so extensive. Still, the authors could improve the Introduction and Discussion by adding references about socioeconomic status and prognosis/outcome in Hodgkin Lymphoma and NHL.
As suggested, we have added one study in reference to the discussion about the difficulties to retrieve information when the analysis is after the study. The SIMONAL study illustrated recently this context.(19)
However, very few studies are available for HL and we have referenced most of them. The social aspects of this disease are really specific specifically regarding the age of patients and we prefer focusing on HL social aspects and limiting the references focusing on NHL or other cancer studies. Moreover, most of the studies are focused on the social aspects of patients after the completion of treatment in survivors while the present study focused on the impact of social aspects on baseline characteristics.
2- The study's objective in the abstract differs from the objective in the introduction.
Our apologies for the discrepancies between the presentation of the objective in the two locations. We have now modified the sentence in the introduction accordingly to those in the introduction of the main text.
3- The retrospective nature of the study and missing information from pts dead at the moment of the analysis prevents a proper analysis of the impact of socio-demographic characteristics on the outcome because this is a critical selection bias. The authors have expressed their concern about this issue in the discussion and argued that the studied population seems to be representative of the AHL2011 cohort, but this is not sufficient to conclude that it is possible to perform an outcome analysis. Therefore, the authors should consider restricting the study to the impact of characteristics on baseline characteristics at diagnosis and prognostic factors, but not the outcome. Another possibility to pursue the outcome analysis is to retrospectively collect occupational position and use it as a proxy of socioeconomic status in patients who died, even though the remaining information from the survey will not be available. This measure is one of the most commonly used indicators of socioeconomic status (Silvia Stringhini*, Cristian Carmeli*, Markus Jokela Published online January 31, 2017 http://dx.doi.org/10.1016/S0140-6736(16)32380-7
We agree that the data analyzed in the present study concern only a subset of long survivors of the AHL2011 study. Unfortunaltely, the occupational position of the dead patient is not extractable from our database, consequently the reviewer’s request cannot be granted.
However, to answer to the reviewer concerns,
- we have divided into several parts the results section to distinguish the correlative analyses of the baseline characteristics and those on patients outcome. Indeed, the title of the final paragraph of the result section dedicated to patients outcome is : “ Relationships of sociodemographic features and Outcome of these long survivor patient”.
- we modified the title of the paper (”Influence of Sociodemographic determinants on the Hodgkin lymphoma baseline characteristics in long survivors patients enrolled in the prospective phase 3 trial AHL2011”).
Reviewer 3 Report
The authors analyze in their study the impact of sociodemographic features on characteristics of Hodgkin lymphoma using data from a multicenter trial (AHL2011). This is an interesting topic, however the study has several limitations. The authors have addressed some of them in the discussion.
The correlation between TMTV and number of inhabitants of zip code is weak (0.29), even if significant. This should be discussed more clearly.
Maximum size of inhabitants appears to be about 80000 in figure 2. This would have excluded patients from most cities.
In figure 2 it is not clear what the short red line between the three black symbols of “TMTV median log” stands for. It appears not to be a regression line, as stated in the figure legend. TMTV median of which groups?
How were rural and urban areas defined in this study?
The term “socially isolated” is not clear. It appears that all patients that are not living in a stable partnership as a couple (married/marriage-like relationship), i.e singles, are defined as “socially isolated”. Most of HL patients are young adults/adolescents, and following the definition of the authors a 20-year old student living with his parents/family would be defined as “socially isolated”. Please explain.
In the discussion (line 261) the authors state that professional activity was associated with less frequent SAE. No data are shown in the results section.
The group of inactive (no professional activity) patients consisted of 66 patients. In the list of socio-professional activities only 15 patients are listed as “without professional activity”.
Are these the unemployed persons? Considering that probably few patients in this young cohort are retired, most patients would fall in the category of “sick leave”.
The association between “sick leave” and higher disease activity (IPS) could appear obvious, and probably the higher disease activity determines the status sick leave, and not the other way round, as the authors suggest.
How would the association between IPD and professional activity be if the category “sick leave” is excluded?
There are numerous language errors that need to be corrected.
Author Response
Reviewer 3
The authors analyze in their study the impact of sociodemographic features on characteristics of Hodgkin lymphoma using data from a multicenter trial (AHL2011). This is an interesting topic, however the study has several limitations. The authors have addressed some of them in the discussion.
1)The correlation between TMTV and number of inhabitants of zip code is weak (0.29), even if significant. This should be discussed more clearly.
We have modulated the impact of this correlation page11:
“This significant correlation remains mitigated (R=0.29) insofar as the numbers of inhabitants of these towns were heterogeneous”
2)Maximum size of inhabitants appears to be about 80000 in figure 2. This would have excluded patients from most cities.
We thank the reviewer to give us the opportunity to clarify this point. The maximum of 80,000 inhabitants is just one format consideration. Indeed, to improve the readability of the graph, we think that is better. However, all towns have been actually included in the calculation.
We have modified the legend of figure 2.
All sizes of towns are considered in the calculation while, for the visualization, the upper limit of the x-axis is 80000).
3)In figure 2 it is not clear what the short red line between the three black symbols of “TMTV median log” stands for. It appears not to be a regression line, as stated in the figure legend. TMTV median of which groups?
We have modified the legend to be more accurate. Indeed, the regression line shows links to the TMTV median of each of the 3 groups.
4)How were rural and urban areas defined in this study?
In the questionnaire, two answers were proposed regarding the type of living area. The definition rural/urban was based on the INSEE (French National Institute of Statistics and Economic Studies) definition with a threshold of 2000 inhabitants including a 200 meters-distance maximum between 2 houses.
We modified the section in methods page 7
“The estimation of the inhabitant's number and urban/rural categories were determined according to the definition of INSEE (French National Institute of Statistics and Economic Studies) with a threshold of 2000 inhabitants. “
Reference19 =
https://www.insee.fr/fr/information/2115018#:~:text=L'unit%C3%A9%20urbaine%20est%20une,population%20dans%20cette%20zone%20b%C3%A2tie.
5) The term “socially isolated” is not clear. It appears that all patients that are not living in a stable partnership as a couple (married/marriage-like relationship), i.e singles, are defined as “socially isolated”. Most of HL patients are young adults/adolescents, and following the definition of the authors a 20-year old student living with his parents/family would be defined as “socially isolated”. Please explain.
We agree with the reviewer to explain better the term. To define marital status, we referred to others papers dealing with HL and social aspects (namely mostly Wang et al., Oncotarget 2016). Our population is actually younger than those in the paper mentioned. But in the questionnaire, the objective of the question about marital status was to distinguish patients living with a partner (married or not or with a family for the youngest) at the diagnosis or alone Thus, 61.9% lived alone which reflects the young age of the trial (median=32 y.o.). As the main goal was to analyze social living in regard to baseline characteristics we assumed that less attention and pressure to visit a physician could be associated to isolated patients. However, patients isolated or not had similar baseline characteristics.
To clarify the description, we added one sentence in the methods section.
‘ In the questionnaire, to taking account of the young age of the trial, the marital status was explored in terms of a partner in the same residency (family, boy/girl friend, spouse) and not restricted to married status only.’
In the results section, we added some details to the section dedicated to analyses of sociodemographic characteristics :
143 (61.9%) out of 232 patients declared to be socially not isolated (i.e. married or in a married life relationship) and 38.1% (n=88) declared to be single (divorced, separated, widowed, student without stable partner), only 2 of whom were widowed, i.e. 0.9% of patients. At diagnosis, the non-isolated patients were older, with a median age of 38.7 years versus 28.7 years in isolated patients (p<0.001).
6) In the discussion (line 261) the authors state that professional activity was associated with less frequent SAE. No data are shown in the results section.
The sentence was present but not properly located. We moved the sentence from the baseline characteristics section in the outcome section.
‘SAEs were less frequent 28.2% (n=46) in active than in inactive patients (42.4% (n=28); p=0.037).’
7) The group of inactive (no professional activity) patients consisted of 66 patients. In the list of socio-professional activities only 15 patients are listed as “without professional activity”.
Are these the unemployed persons? Considering that probably few patients in this young cohort are retired, most patients would fall in the category of “sick leave”.
As suggested, we detailed the counts of each context.
Amongst patients without professional activity at diagnosis, 3 were retired, 24 were unemployed, and 39 in sick leave. The 15 patients listed in the text in the section describing the type of work are patients with unstable activity and they were wrongly entitled.
We have modified it accordingly in the results section on page 9.
‘Overall, 163 patients (71.2%) had professional activity, and 66 (28.8%) were inactive owing to unemployment (n=24), retirement (n=3), or sick leave (n=39).’
8) The association between “sick leave” and higher disease activity (IPS) could appear obvious, and probably the higher disease activity determines the status sick leave, and not the other way round, as the authors suggest. How would the association between IPS and professional activity be if the category “sick leave” is excluded?
Due to the very low number of patients in each category, specifically for the high IPS and inactive group (n=9) ,the link is no longer significant when we exclude sick leave patients (p=0.144) but 18/102 (18%) of patients with high IPS had no professional activity vs 9/88 (10%) patients with low IPS. We have mentioned the results accordingly page7/8.
9) There are numerous language errors that need to be corrected.
We have improved the language thanks to the proofreading by native English medical writer.
Round 2
Reviewer 2 Report
I think the article can be accepted in its present form.
Reviewer 3 Report
The authors addressed all queries